# PolyLoss: A Polynomial Expansion Perspective of Classification Loss Functions

**Zhaoqi Leng**[1], **Mingxing Tan**[1], **Chenxi Liu**[1], **Ekin Dogus Cubuk**[2], **Xiaojie Shi**[2],
**Shuyang Cheng**[1],**Dragomir Anguelov**[1]
[1]Waymo LLC    [2]Google LLC
```
{lengzhaoqi, tanmingxing, cxliu, shuyangcheng, dragomir}@waymo.com
{cubuk, xiaojies}@google.com
```

## ABSTRACT

Cross-entropy loss and focal loss are the most common choices when training deep neural networks for classification problems. Generally speaking, however, a good loss function *can* take on much more flexible forms, and *should* be tailored for different tasks and datasets. Motivated by how functions can be approximated via Taylor expansion, we propose a simple framework, named *PolyLoss*, to view and design loss functions as a linear combination of polynomial functions. Our PolyLoss allows the importance of different polynomial bases to be easily adjusted depending on the targeting tasks and datasets, while naturally subsuming the aforementioned cross-entropy loss and focal loss as special cases. Extensive experimental results show that the optimal choice within the PolyLoss is indeed dependent on the task and dataset. Simply by introducing one extra hyperparameter and adding one line of code, our *Poly-1* formulation outperforms the cross-entropy loss and focal loss on 2D image classification, instance segmentation, object detection, and 3D object detection tasks, sometimes by a large margin.

| Task | ImageNet classification | | COCO det. and seg. | | Waymo Open Dataset 3D detection | | | |
| --- | --- | --- | --- | --- | --- | --- | --- | --- |
| Default loss | Cross-entropy | | Cross-entropy | | Focal loss | | | |
| Model | ENetV2-L(21K) | ENetV2-L(1K) | Mask R-CNN | | PointPillars Car | PointPillars Ped | RSN Car | RSN Ped |
| Baseline | 45.8 | 86.8 | 47.2 | 42.3 | 63.3 | 68.9 | 78.4 | 79.4 |
| PolyLoss | **46.4** (+0.6) | **87.2** (+0.4) | **49.7** (+2.5) | **44.4** (+2.1) | **63.7** (+0.4) | **69.6** (+0.7) | **78.9** (+0.5) | **80.2** (+0.8) |

Table 1: **PolyLoss outperforms cross-entropy and focal loss on various models and tasks.** Results are for the simplest Poly-1, which has only a single hyperparameter. On ImageNet (Deng et al., 2009), our PolyLoss improves both pretraining and finetuning for the recent EfficientNetV2 (Tan & Le, 2021); on COCO (Lin et al., 2014), PolyLoss improves both 2D detection and segmentation AR for Mask-RCNN (He et al., 2017); on Waymo Open Dataset (WOD) (Sun et al., 2020), PolyLoss improves 3D detection AP for the widely used PointPillars (Lang et al., 2019) and the very recent Range Sparse Net (RSN) (Sun et al., 2021). Details are in Table 4, 5, 7.

## 1 INTRODUCTION

Loss functions are important in training neural networks. In principle, a loss function could be any (differentiable) function that maps predictions and labels to a scalar. Therefore, designing a good loss function is generally challenging due to its large design space, and designing a universal loss function that works across different tasks and datasets is even more challenging: for example, L1 / L2 losses are commonly used for regression tasks, but they are rarely used for classification tasks; focal loss is often used to alleviate the overfitting issue of cross-entropy loss for imbalanced object detection datasets (Lin et al., 2017), but it is not shown to consistently help other tasks. Many recent works have also explored new loss functions via meta-learning, ensembling or compositing different losses (Hajiabadi et al., 2017; Xu et al., 2018; Gonzalez & Miikkulainen, 2020b;a; Li et al., 2019).

In this paper, we propose *PolyLoss*: a novel framework for understanding and designing loss functions. Our key insight is to decompose commonly used classification loss functions, such as cross-entropy loss and focal loss, into a series of weighted polynomial bases. They are decomposed in the form of $\sum_{j=1}^{\infty} \alpha_j (1 - P_t)^j$, where $\alpha_j \in \mathbb{R}^+$ is the polynomial coefficient and $P_t$ is the prediction probability of the target class label. Each polynomial base $(1 - P_t)^j$ is weighted by a corresponding polynomial coefficient $\alpha_j$, which enables us to easily adjust the importance of different bases for different applications. When $\alpha_j = 1/j$ for all $j$, our PolyLoss becomes equivalent to the commonly used cross-entropy loss, but this coefficient assignment may not be optimal.

Our study shows that, in order to achieve better results, it is necessary to adjust polynomial coefficients $\alpha_j$ for different tasks and datasets. Since it is impossible to adjust an infinite number of $\alpha_j$, we explore various strategies with a small degree of freedom. Perhaps surprisingly, we observe that simply adjusting the single polynomial coefficient for the leading polynomial, which we denote $L_{\text{Poly-1}}$, is sufficient to achieve significant improvements over the commonly used cross-entropy loss and focal loss. Overall, our contribution can be summarized as:

- **Insights on common losses**: We propose a unified framework, named *PolyLoss*, to rethink and redesign loss functions. This framework helps to explain cross-entropy loss and focal loss as two special cases of the PolyLoss family (by *horizontally* shifting polynomial coefficients), which was not recognized before. This new finding motivates us to investigate new loss functions that *vertically* adjust polynomial coefficients, shown in Figure 1.
- **New loss formulation:** We evaluate different ways of *vertically* manipulating polynomial coefficients to simplify the hyperparameters search space. We propose a simple and effective **Poly-1** loss formulation which only introduces one hyperparameter and one line of code.
- **New findings:** We identify that focal loss, though effective for many detection tasks, is suboptimal for the imbalanced ImageNet-21K. We find the leading polynomial contributes to a large portion of the gradient during training, and its coefficient correlates to the prediction confidence $P_t$. In addition, we provide an intuitive explanation on how to leverage this correlation to design good PolyLoss tailored to imbalanced datasets.
- **Extensive experiments:** We evaluate our PolyLoss on different tasks, models, and datasets. Results show PolyLoss consistently improves the performance on all fronts, summarized in Table 1, which includes the state-of-the-art classifiers EfficientNetV2 and detectors RSN.

## 2 RELATED WORK

Cross-entropy loss is used in popular and current state-of-the-art models for perception tasks such as classification, detection and semantic segmentation (Tan & Le, 2021; He et al., 2017; Zoph et al., 2020; Tao et al., 2020). Various losses are proposed to improve cross-entropy loss (Lin et al., 2017; Law & Deng, 2018; Cui et al., 2019; Zhao et al., 2021). Unlike prior works, the goal of this paper is to provide a unified framework for systematically designing a better classification loss function.

**Loss for class imbalance** Training detection models, especially single-stage detectors, is difficult due to class imbalance. Common approaches such as hard example mining and reweighing are developed to address the class imbalance issue (Sung, 1996; Viola & Jones, 2001; Felzenszwalb et al., 2010; Shrivastava et al., 2016; Liu et al., 2016; Bulo et al., 2017). As one of these approaches, focal loss is designed to mitigate the class imbalance issue by focusing on the hard examples and is used to train state-of-the-art 2D and 3D detectors (Lin et al., 2017; Tan et al., 2020; Du et al., 2020; Shi et al., 2020; Sun et al., 2021). In our work, we found that focal loss is suboptimal for the imbalanced ImageNet-21K. Using the PolyLoss framework, we discover a better loss function, which performs the opposite role of focal loss. We further provide intuitive understanding of why it is important to design different loss functions tailored to different imbalanced datasets using the PolyLoss framework.

**Robust loss to label noise** Another direction of research is to design loss functions that are robust to label noise (Ghosh et al., 2015; 2017; Zhang & Sabuncu, 2018; Wang et al., 2019; Oksuz et al., 2020; Menon et al., 2019). A commonly used approach is to incorporate noise robust loss function such as Mean Absolute Error (MAE) into cross-entropy loss. In particular, Taylor cross entropy loss is proposed to unify MAE and cross-entropy loss by expanding the cross-entropy loss in $(1 - P_t)^j$ polynomial bases (Feng et al., 2020). By truncating the higher-order polynomials, they show truncated cross-entropy loss function is closer to MAE, which is more robust to label noise on datasets with synthetic label noise. In contrast, our PolyLoss provides a more general framework to design loss functions for different datasets by manipulating polynomial coefficients, which includes dropping higher-order polynomials proposed in Feng et al. (2020). Our experiments in subsection 4.1 show the loss proposed in Feng et al. (2020) performs *worse* than cross-entropy loss on the clean ImageNet dataset.

**Learned loss functions** Several recent works demonstrate learning the loss function during training via gradient descent or meta learning (Hajiabadi et al., 2017; Xu et al., 2018; Gonzalez & Miikkulainen, 2020a; Li et al., 2019; 2020). Notably, TaylorGLO utilizes CMA-ES to optimize multivariate Taylor parameterization of a loss function and learning rate schedule during training (Hansen & Ostermeier, 1996; Gonzalez & Miikkulainen, 2020b). Due to the search space scale with the order

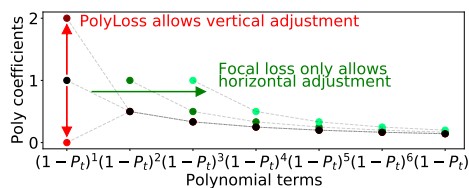

Figure 1: **Unified view of cross-entropy loss, focal loss, and PolyLoss**. PolyLoss $\sum_{j=1}^{\infty} \alpha_j (1 - P_t)^j$ is a more general framework, where $P_t$ stands for prediction probability of the target class. Left: Polyloss is more flexible: it can be steeper (deep red) than cross-entropy loss (black) or flatter (light red) than focal loss (green). Right: Polynomial coefficients of different loss functions in the bases of $(1 - P_t)^j$, where $j \in \mathbb{Z}^+$. Black dash lines are drawn to show the trend of polynomial coefficients. In the PolyLoss framework, focal loss can only shift the polynomial coefficients *horizontally* (green arrow), see Equation 2, whereas the proposed PolyLoss framework is more general, which also allows *vertical* adjustment (red arrows) of the polynomial coefficient for each polynomial term.

of polynomials, the paper demonstrates that using the third-order parameterization (8 parameters), the learned loss function schedule outperforms cross-entropy loss on 10-class classification problems. Our paper (Figure 2a), on the other hand, shows for 1000-class classification tasks, hundreds of polynomials are needed. This results in a prohibitively large search space. Our proposed Poly-1 formulation mitigates the challenge of the large search space and do not rely on advanced black-box optimization algorithms. Instead, we show a simple grid search over one hyperparameter can lead to significant improvement on all tasks that we investigate.

## 3 POLYLOSS

PolyLoss provides a framework for understanding and improving the commonly used cross-entropy loss and focal loss, visualized in Figure 1. It is inspired from the Taylor expansion of cross-entropy loss (Equation 1) and focal loss (Equation 2) in the bases of $(1 - P_t)^j$:

$$L_{\text{CE}} = -\log(P_t) = \sum_{j=1}^{\infty} 1/j(1 - P_t)^j = (1 - P_t) + 1/2(1 - P_t)^2 ... \tag{1}$$

$$L_{\text{FL}} = -(1 - P_t)^\gamma \log(P_t) = \sum_{j=1}^{\infty} 1/j(1 - P_t)^{j+\gamma} = (1 - P_t)^{1+\gamma} + 1/2(1 - P_t)^{2+\gamma} ... \tag{2}$$

where $P_t$ is the model's prediction probability of the target ground-truth class.

**Cross-entropy loss as PolyLoss** Using the gradient descent method to optimize the cross-entropy loss requires taking the gradient with respect to $P_t$. In the PolyLoss framework, an interesting observation is that the coefficients $1/j$ exactly cancel the $j$th power of the polynomial bases, see Equation 1. Thus, the gradient of cross-entropy loss is simply the sum of polynomials $(1 - P_t)^j$, shown in Equation 3.

$$-\frac{dL_{\text{CE}}}{dP_t} = \sum_{j=1}^{\infty} (1 - P_t)^{j-1} = 1 + (1 - P_t) + (1 - P_t)^2 ... \tag{3}$$

The polynomial terms in the gradient expansion capture different sensitivity with respect to $P_t$. The leading gradient term is 1, which provides a constant gradient regardless of the value of $P_t$. On the contrary, when $j \gg 1$, the $j$th gradient term is strongly suppressed when $P_t$ gets closer to 1.

**Focal loss as PolyLoss** In the PolyLoss framework, Equation 2, it is apparent that the focal loss simply shifts the power $j$ by the power of a modulating factor $\gamma$. This is equivalent to *horizontally* shifting all the polynomial coefficients by $\gamma$ as shown in Figure 1. To understand the focal loss from a gradient prospective, we take the gradient of the focal loss (Equation 2) with respect to $P_t$:

$$-\frac{dL_{\text{FL}}}{dP_t} = \sum_{j=1}^{\infty} (1 + \gamma/j)(1 - P_t)^{j+\gamma-1} = (1 + \gamma)(1 - P_t)^\gamma + (1 + \gamma/2)(1 - P_t)^{1+\gamma} ... \tag{4}$$

For a positive $\gamma$, the gradient of focal loss drops the constant leading gradient term, 1, in the cross-entropy loss, see Equation 3. As discussed in the previous paragraph, this constant gradient term causes the model to emphasize the majority class, since its gradient is simply the total number of

| | Polynomial expansion in the basis of $(1 - P_t)$ | Loss |
|---|---|---|
| Cross-entropy loss | $(1 - P_t) + 1/2(1 - P_t)^2 + ... + 1/N(1 - P_t)^N + 1/(N+1)(1 - P_t)^{N+1} + ...$ | $L_{CE} = -\log(P_t)$ |
| Drop poly. (Sec 4.1) | $(1 - P_t) + 1/2(1 - P_t)^2 + ... + 1/N(1 - P_t)^N$ (drop the remaining terms) | $L_{Drop} = L_{CE} - \sum_{j=N}^{\infty} 1/j(1 - P_t)^j$ |
| Poly-N (Sec 4.2) | $(\epsilon_1 + 1)(1 - P_t) + ... + (\epsilon_N + 1/N)D_t^N + 1/(N+1)(1 - P_t)^{N+1} + ...$ | $L_{Poly-N} = L_{CE} + \sum_{j=1}^{N} \epsilon_j(1 - P_t)^i$ |
| Poly-1 (Sec 4.3) | $(\epsilon_1 + 1)(1 - P_t) + 1/2(1 - P_t)^2 + ... + 1/N(1 - P_t)^N + 1/(N+1)(1 - P_t)^{N+1} + ...$ | $L_{Poly-1} = L_{CE} + \epsilon_1(1 - P_t)$ |

Table 2: **Comparing different losses in the PolyLoss framework.** Dropping higher order polynomial, proposed in prior works, truncates all higher order $(N + 1 \to \infty)$ polynomial terms. We propose Poly-N loss, which perturbs the leading N polynomial coefficients. Poly-1 is the final loss formulation, which further simplifies Poly-N and only requires a simple grid search over one hyperparameter. The differences compared to cross-entropy loss are highlighted in red.

examples for each class. By shifting the power of all the polynomial terms by $\gamma$, the first term then becomes $(1 - P_t)^\gamma$, which is suppressed by the power of $\gamma$ to avoid overfitting to the already confident (meaning $P_t$ close to 1) majority class. More details are shown in section 12.

**Connection to regression and general form** Representing the loss function in the PolyLoss framework provides an intuitive connection to regression. For classification tasks where $y = 1$ is the effective probability of the ground-truth label, the polynomial bases $(1 - P_t)^j$ can be expressed as $(y - P_t)^j$. Thus both cross-entropy loss and focal loss can be interpreted as a weighted ensemble of distances between the prediction and label to the $j$th power. However, a fundamental question in those losses: *Are the coefficients in front of the regression terms optimal?*

In general, PolyLoss is a monotone decreasing function[1] on $[0, 1]$ which can be expressed as $\sum_{j=1}^{\infty} \alpha_j(1 - P_t)^j$ and provides a flexible framework to adjust each coefficient[2]. PolyLoss can be generalized to non-integer $j$, but for simplicity we only focus on integer power ($j \in \mathbb{Z}^+$) in this paper. In the next section, we investigate several strategies on designing better loss functions in the PolyLoss framework via manipulating $\alpha_j$.

## 4 UNDERSTANDING THE EFFECT OF POLYNOMIAL COEFFICIENTS

In the previous section, we established the PolyLoss framework and showed that cross-entropy loss and focal loss simply correspond to different polynomial coefficients, where focal loss *horizontally* shifts the polynomial coefficients of cross-entropy loss.

In this section, we propose the final loss formulation **Poly-1**. We study in depth how *vertically* adjusting polynomial coefficients, shown in Figure 1, may affect training. Specifically, we explore three different strategies in assigning polynomial coefficients: dropping higher-order terms; adjusting multiple leading polynomial coefficients; and adjusting the first polynomial coefficient, summarized in Table 2. We find adjusting the first polynomial coefficient (Poly-1 formulation) leads to *maximal* gain while requiring *minimal* code change and hyperparameter tuning.

In these explorations, we experiment with 1000-class ImageNet (Deng et al., 2009) classification. We abbreviate it as ImageNet-1K to differentiate it from the full version, which contains 21K classes. We use ResNet-50 (He et al., 2016) and its training hyperparameters without modification.[3]

### 4.1 $L_{Drop}$: REVISITING DROPPING HIGHER-ORDER POLYNOMIAL TERMS

Prior works (Feng et al., 2020; Gonzalez & Miikkulainen, 2020b) have shown dropping the higher-order polynomials and tuning the leading polynomials can improve model robustness and performance. We adopt the same loss formulation $L_{Drop} = \sum_{j=1}^{N} 1/j(1 - P_t)^j$, as in Feng et al. (2020), and compare their performance with the baseline cross-entropy loss on ImageNet-1K. As shown in Figure 2a, we need to sum up more than 600 polynomial terms to match the accuracy of cross-entropy loss. Notably, removing higher-order polynomials cannot simply be interpreted as adjusting the learning rate. To verify this, Figure 2b compares the performance for different learning rates with various cutoffs: no matter we increase or decrease the learning rate from the original value of 0.1, the accuracy worsens. Additional hyperparameter tuning is shown in section 9.

---

[1]We only consider the case all $\alpha_j \geq 0$ in this paper for simplicity. There exist monotone decreasing functions on $[0, 1]$ with some $\alpha_j$ negative, for example $\sin(1 - P_t) = \sum_{j=0}^{\infty} (-1)^j/(2j + 1)!(1 - P_t)^{2j+1}$.

[2]To ensure series converges, we require $1/\limsup_{j\to\infty} \sqrt[j]{|\alpha_j|} \geq 1$ for $P_t$ in $(0, 1]$. For $P_t = 0$ we don't require point-wise convergence; in fact cross-entropy and focal loss both go to $+\infty$.

[3]Code at https://github.com/tensorflow/tpu/tree/master/models/official/

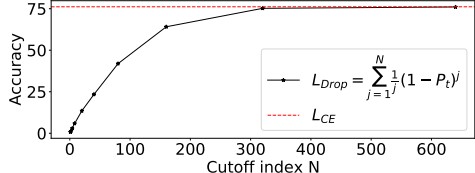

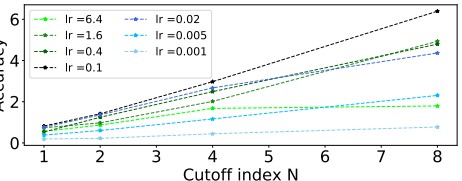

(a) Truncating the infinite sum of polynomials in cross-entropy loss to $N$ reduces accuracy.

(b) Adjusting the learning rate (default 0.1) of $L_{\text{Drop}}$ does not improve the classification accuracy.

Figure 2: **Training ResNet-50 on ImageNet-1K requires hundreds of polynomial terms to reproduce the same accuracy as cross-entropy loss.**

To understand why higher-order terms are important, we consider the residual sum after removing the first $N$ polynomial terms from cross-entropy loss: $R_{\text{N}} = L_{\text{CE}} - L_{\text{Drop}} = \sum_{j=N+1}^{\infty} 1/j(1-P_t)^j$.

**Theorem 1.** *For any small $\zeta > 0$, $\delta > 0$ if $N > \log_{1-\delta}(\zeta \cdot \delta)$, then for any $p \in [\delta, 1]$, we have $|R_N(p)| < \zeta$ and $|R'_N(p)| < \zeta$. (Proof in section 7)*

Hence, taking a large $N$ is necessary to ensure $L_{\text{Drop}}$ is uniformly close to $L_{\text{CE}}$ in the perspectives of loss and loss derivative on $[\delta, 1]$. For a fixed $\zeta$, as $\delta$ approaches 0, $N$ grows rapidly. Our experimental results align with the theorem. The higher-order ($j > N + 1$) polynomials play an important role during the early stages of training, where $P_t$ is typically close to zero. For example, when $P_t \sim 0.001$, according to Equation 3, the coefficient of the 500th term's gradient is $0.999^{499} \sim 0.6$, which is fairly large. Different from aforementioned prior works, our results show that we cannot easily reduce the number of polynomial coefficients $\alpha_j$ by excluding the higher-order polynomials.

Dropping higher order polynomials is equivalent to pushing all the higher order ($j > N+1$) polynomial coefficients $\alpha_j$ vertically to zero in the PolyLoss framework. Since simply setting coefficients to zero is suboptimal for training ImageNet-1K, in the following sections, we investigate how to manipulate polynomial coefficient beyond setting them to zero in the PolyLoss framework. In particular, *we aim to propose a simple and effective loss function that requires minimal tuning.*

## 4.2 $L_{\text{POLY-N}}$: PERTURBING LEADING POLYNOMIAL COEFFICIENTS

In this paper, we propose an alternative way of designing a new loss function in the PolyLoss framework, where we adjust the coefficients of each polynomial. In general, there are infinitely many polynomial coefficients $\alpha_j$ need to be tuned. Thus, it is infeasible to optimize the most general loss:

$$L_{\text{Poly}} = \alpha_1(1 - P_t) + \alpha_2(1 - P_t)^2 + ... + \alpha_N(1 - P_t)^N + ... = \sum_{j=1}^{\infty} \alpha_j(1 - P_t)^j \qquad (5)$$

The previous section (subsection 4.1) has shown that hundreds of polynomials are required in training to do well on tasks such as ImageNet-1K classification. If we naively truncate the infinite sum in Equation 5 to the first few hundreds terms, tuning coefficients for so many polynomials still results in a prohibitively large search space. In addition, collectively tuning many coefficients also does not outperform cross-entropy loss, details in section 10.

To tackle this challenge, we propose to *perturb* the leading polynomial coefficients in cross-entropy loss, while keeping the rest the same. We denote the proposed loss formulation as Poly-N, where N stands for the number of leading coefficients that will be tuned.

$$L_{\text{Poly-N}} = \underbrace{(\epsilon_1 + 1)(1 - P_t) + ... + (\epsilon_N + 1/N)(1 - P_t)^N}_{\text{perturbed by } \epsilon_j} + \underbrace{1/(N+1)(1 - P_t)^{N+1} + ...}_{\text{same as } L_{\text{CE}}}$$

$$= -\log(P_t) + \sum_{j=1}^{N} \epsilon_j(1 - P_t)^j \qquad (6)$$

Here, we replace the $j$th polynomial coefficient in cross-entropy loss $1/j$ with $1/j + \epsilon_j$, where $\epsilon_j \in [-1/j, \infty)$ is the perturbation term. This allows us to pinpoint the first $N$ polynomials without the need to worry about the infinitely many higher-order ($j > N + 1$) coefficients, as in Equation 5.

| | CE loss | N=1 | N=2 | N=3 |
|---|---|---|---|---|
| N-dim. grid search | 76.3 | 76.7 | 76.8 | – |
| Greedy grid search | 76.3 | 76.7 | 76.7 | 76.7 |

Table 3: $L_{\text{Poly-N}}$ **outperforms cross-entropy loss on ImageNet-1K.**

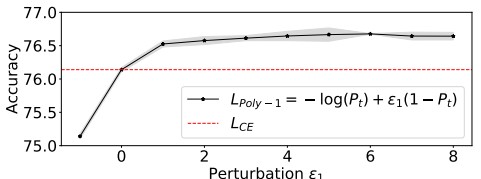 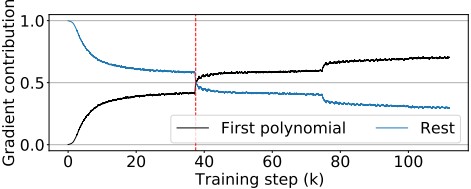

(a) PolyLoss family $L_{\text{Poly-1}} = -\log(P_t) + \epsilon_1(1 - P_t)$, where $\epsilon_1 \in \{-1, 0, 1, \ldots, 8\}$.

(b) Percentage of gradient from the first polynomial versus the rest (infinitely many) polynomials.

Figure 3: **The first polynomial plays an important role for training ResNet-50 on ImageNet-1K.** (a) Increasing the coefficient of the first polynomial term ($\epsilon_1 > 0$) consistently improves the ResNet50 prediction accuracy. Red dash line shows the accuracy when using cross-entropy loss. Mean and stdev of three runs are plotted. (b) The first polynomial $(1 - P_t)$ contributes more than half of the cross-entropy gradient at the last 65% of the training steps, which highlights the importance of tuning the first polynomial. The red dash line shows the crossover.

Table 3 shows $L_{\text{Poly-N}}$ outperforms the baseline cross-entropy loss accuracy. We explore N-dimensional grid search and greedy grid search of $\epsilon_j$ in $L_{\text{Poly-N}}$ up to $N = 3$ and find that simply adjusting the coefficient of the first polynomial ($N = 1$) leads to better classification accuracy. Performing 2D grid search ($N = 2$) can further boost the accuracy. However, the additional gain is small (+0.1) compared to adjusting only the first polynomial (+0.4).

### 4.3 $L_{\text{POLY-1}}$: SIMPLE AND EFFECTIVE

As shown in the previous section, we find tuning the first polynomial term leads to the most significant gain. In this section, we further simplify the Poly-N formulation and focus on evaluating Poly-1, where only the first polynomial coefficient in cross-entropy loss is modified.

$$L_{\text{Poly-1}} = (1 + \epsilon_1)(1 - P_t) + 1/2(1 - P_t)^2 + \ldots = -\log(P_t) + \epsilon_1(1 - P_t) \tag{7}$$

We study the effect of different first term scaling on the accuracy and observe that increasing the first polynomial coefficient can systematically increase the ResNet-50 accuracy, as shown in Figure 3a. This result suggests that the cross-entropy loss is suboptimal in terms of polynomial coefficient values, and increasing the first polynomial coefficient leads to consistent improvement, which is comparable to other training techniques (section 11).

Figure 3b shows the leading polynomial contributes to more than half of the cross-entropy gradient during training for the majority of the time, which highlights the significance of the first polynomial term $(1 - P_t)$ compared to the rest of the infinite many terms. Therefore, in the remaining of the paper, we adopt the form of $L_{\text{Poly-1}}$ and primarily focus on adjusting the leading polynomial coefficient. As is evident from Equation 7, it only modifies the original loss implementation by a single line of code (adding a $\epsilon_1(1 - P_t)$ term on top of cross-entropy loss).

Note that, all the training hyperparameters are optimized for cross-entropy loss. Even so, a simple grid search on the first polynomial coefficients in the Poly-1 formulation significantly increases the classification accuracy. We find optimizing other hyperparameters for $L_{\text{Poly-1}}$ leads to higher accuracy, and show more details in section 8.

## 5 EXPERIMENTAL RESULTS

In this section, we compare our PolyLoss against the commonly used cross-entropy loss and focal loss on various tasks, models, and datasets. For the following experiments, we adopt the default training hyperparameters in the public repositories without any tuning. Nevertheless, Poly-1 formulation leads to consistent advantage over default loss functions at the cost of a simple grid search.

### 5.1 $L_{\text{POLY-1}}$ IMPROVES 2D IMAGE CLASSIFICATION ON IMAGENET

Image classification is a fundamental problem in computer vision, and progress on image classification has led to progress on many related computer vision tasks. In terms of the network architecture, in addition to the ResNet-50 already used in section 4, we also experiment with the state-of-the-art EfficientNetV2 (Tan & Le, 2021). We use the ImageNet settings in (Tan & Le, 2021) except for replacing the original cross-entropy loss with our PolyLoss $L_{Poly-1}$ with different values of $\epsilon_1$. In terms of the dataset, in addition to the ImageNet-1K dataset already used in section 4, we also consider ImageNet-21K, which has about 13M training images with 21,841 classes. We will study both the ImageNet-21K pretraining results and the ImageNet-1K finetuning results.

Pretraining EfficientNetV2-L on ImageNet-21K, then finetuning it on ImageNet-1K can improve classification accuracy (Tan & Le, 2021). Here, we follow the same pretraining and finetuning schedule as reported in Tan & Le (2021) without modification[4] but replace the cross-entropy loss with $L_{\text{Poly-1}} = -\log(P_t) + \epsilon_1(1 - P_t)$. We reserve 25,000 images from the training set as *minival* to search the optimal $\epsilon_1$.

**Pretraining on ImageNet-21K** Figure 4 highlights the importance of using tailored loss function when pretraining model on ImageNet-21K dataset. A simple grid search over $\epsilon_1 \in \{0, 1, 2, \ldots, 7\}$ in $L_{\text{Poly-1}}$ without changing other default hyperparameters leads to around 1% accuracy gain for all SOTA EfficientNetV2 models with different sizes. The accuracy improvement of using a better loss function nearly matches the improvement of scaling up the model architecture (S to M and M to L).

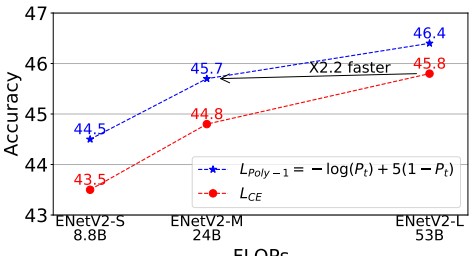

Figure 4: **PolyLoss improves EfficientNetV2 family on the speed-accuracy Pareto curve.** Validation accuracy of EfficientNetV2 models pretrained on ImageNet-21K are plotted. PolyLoss outperforms cross-entropy loss with about ×2 speed-up.

Surprisingly, see Figure 5a, increasing the weight of the leading polynomial coefficient improves the accuracy of pretraining on ImageNet-21K (+0.6), whereas reducing it lowers the accuracy (-0.9). Setting $\epsilon_1 = -1$ truncates the leading polynomial term in the cross-entropy loss (Equation 1), which is similar to having a focal loss with $\gamma = 1$ (Equation 2). However, the opposite change, where $\epsilon_1 > 0$, improves the accuracy on the imbalanced ImageNet-21K.

We hypothesize the prediction of the imbalanced ImageNet-21K is not confident enough ($P_t$ is small), and using positive $\epsilon_1$ PolyLoss leads to more confident predictions. To validate our hypothesis, we plot $P_t$ as a function of training steps in Figure 5b. We observe that $\epsilon_1$ directly controls the mean $P_t$ over all classes. Using positive $\epsilon_1$ PolyLoss leads to more confident prediction (higher $P_t$). On the other hand, negative $\epsilon_1$ PolyLoss lowers the confidence.

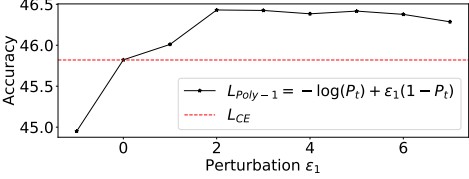

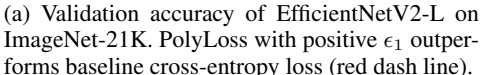

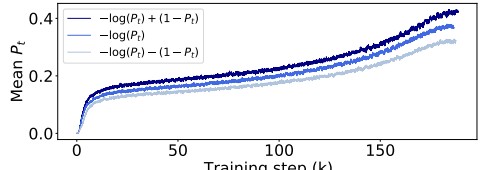

(a) Validation accuracy of EfficientNetV2-L on ImageNet-21K. PolyLoss with positive $\epsilon_1$ outperforms baseline cross-entropy loss (red dash line).

(b) Positive $\epsilon_1 = 1$ (dark) increases the prediction confidence, while negative $\epsilon_1 = -1$ (light) decreases the prediction confidence.

Figure 5: **PolyLoss improves EfficientNetV2-L by increasing prediction confidence $P_t$.**

**Fine tuning on ImageNet-1K** After pretraining on ImageNet-21K, we take the EfficientNetV2-L checkpoint and finetune it on ImageNet-1K, using the same procedure as Tan & Le (2021) except for replacing the original cross-entropy loss with the Poly-1 formulation. PolyLoss improves the finetuning accuracy by 0.4%, advancing the ImageNet-1K top-1 accuracy from 86.8% to 87.2%.

| EfficientNetV2-L | $L_{\text{CE}}$ | $L_{\text{Poly-1}}$ | Improv. |
|---|---|---|---|
| ImageNet-21K | 45.8 | **46.4** | **+0.6** |
| ImageNet-1K | 86.8 | **87.2** | **+0.4** |

Table 4: **PolyLoss improves classification accuracy on ImageNet *validation set*.** We set $\epsilon_1 = 2$ for both.

### 5.2 $L_{\text{POLY-1}}$ IMPROVES 2D INSTANCE SEGMENTATION AND OBJECT DETECTION ON COCO

Instance segmentation and object detection require localizing objects in an image in addition to recognizing them: the former in the form of arbitrary shapes and the latter in the form of bounding boxes. For both instance segmentation and object detection, we use the popular COCO (Lin et al., 2014) dataset, which contains 80 object classes. We choose Mask R-CNN (He et al., 2017) as the representative model for instance segmentation and object detection. These models optimize multiple losses, e.g. $L_{\text{MaskRCNN}} = L_{\text{cls}} + L_{\text{box}} + L_{\text{mask}}$. For the following experiments, we only replace the $L_{\text{cls}}$ with PolyLoss and leave other losses intact. Results are summarized in Table 5.

---

[4] Code at https://github.com/google/automl/tree/master/efficientnetv2

| | Loss | Box | | Mask | |
|---|---|---|---|---|---|
| | | AP | AR | AP | AR |
| Mask R-CNN $L_{\text{CE}}$ | $-\log(P_t)$ | $35.0 \pm 0.09$ | $47.2 \pm 0.16$ | $31.3 \pm 0.09$ | $42.3 \pm 0.02$ |
| Mask R-CNN $L_{\text{Poly-1}}$ | $-\log(P_t) - (1 - P_t)$ | $\mathbf{35.3 \pm 0.12}$ | $\mathbf{49.7 \pm 0.07}$ | $\mathbf{31.6 \pm 0.11}$ | $\mathbf{44.4 \pm 0.07}$ |
| Improvement | - | +0.3 | +2.5 | +0.3 | +2.1 |

Table 5: **PolyLoss improves detection results on COCO *validation set*.** Bounding box and instance segmentation mask average-precision (AP) and average-recall (AR) are reported for Mask R-CNN model with a ResNet-50 backbone. Mean and stdev of three runs are reported.

**Reducing the leading polynomial coefficient improves Mask R-CNN AP and AR.** In training Mask R-CNN, we use the training schedule optimized for cross-entropy loss,[5] and replace the cross-entropy loss with $L_{Poly-1} = -\log(P_t) + \epsilon_1(1 - P_t)$ for the classification loss $L_{cls}$, where $\epsilon_1 \in \{-1.0, -0.8, -0.6, -0.4, -0.2, 0, 0.5, 1.0\}$. We ensure the leading coefficient is positive, i.e. $\epsilon_1 \geq -1$. Our results in Figure 6a show systematic improvements of box AP, box AR, mask AP, and mask AR as we reduce the weight of the first polynomial by using negative $\epsilon_1$ values. Note that Poly-1 ($\epsilon = -1$) not only improves AP but also significantly increases AR, shown in Table 5.

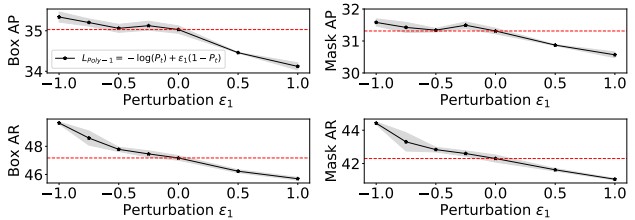
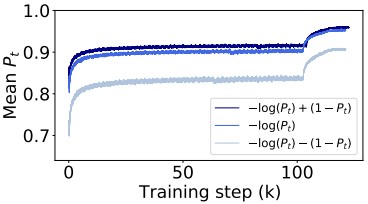

(a) Bound box AP, AR and Mask AP, AR increase as $\epsilon_1$ decreases. Negative $\epsilon_1$ outperforms cross-entropy loss (red dash line).

(b) Negative $\epsilon_1 = -1$ (light) reduces the overconfident prediction $P_t$.

Figure 6: **PolyLoss improves Mask R-CNN by lowering overconfident predictions.** Mean and stdev of three runs are plotted.

**Tailoring loss function to datasets and tasks is important.** ImageNet-21K and COCO are both imbalanced but the optimal $\epsilon$ for PolyLoss are opposite in sign, i.e. $\epsilon = 2$ for ImageNet-21K classification and $\epsilon = -1$ for Mask R-CNN detection. We plot the $P_t$ of the Mask R-CNN classification head and found the original prediction is overly confident ($P_t$ is close to 1) on the imbalanced COCO dataset, thus using a negative $\epsilon$ lowers the prediction confidence, as shown in Figure 6b. This effect is similar to label smoothing (Szegedy et al., 2016) and confidence penalty (Pereyra et al., 2017), but unlike those methods, as long as $0 > \epsilon > -1$, PolyLoss lowers the gradients of overconfident predictions but will not encourage incorrect predictions or directly penalize prediction confidence.

### 5.3 $L_{\text{POLY-1}}$ IMPROVES 3D OBJECT DETECTION ON WAYMO OPEN DATASET

| | Polynomial expansion in the basis of $(1 - P_t)$ | Loss |
|---|---|---|
| Focal loss | $(1 - P_t)^{\gamma+1} + 1/2(1 - P_t)^{\gamma+2} + 1/3(1 - P_t)^{\gamma+3} + ...$ | $L_{\text{FL}} = -(1 - P_t)^\gamma \log(P_t)$ |
| Poly-1 (PointPillars) | $(\epsilon_1 + 1)(1 - P_t)^{\gamma+1} + 1/2(1 - P_t)^{\gamma+2} + 1/3(1 - P_t)^{\gamma+3} + ...$ | $L_{\text{Poly-1}}^{\text{FL}} = L_{\text{FL}} + \epsilon_1(1 - P_t)^{\gamma+1}$ |
| Poly-1* (RSN) | (drop first) $(1/2 + \epsilon_2)(1 - P_t)^{\gamma+2} + 1/3(1 - P_t)^{\gamma+3} + ...$ | $L_{\text{Poly-1*}}^{\text{FL}} = L_{\text{FL}} - (1 - P_t)^{\gamma+1} + \epsilon_2(1 - P_t)^{\gamma+2}$ |

Table 6: **PolyLoss vs. focal loss for 3D detection models.** Differences are highlighted in red. We found the best Poly-1 for PointPillars is $\epsilon_1 = -1$, which is equivalent to dropping the first term. Therefore, for RSN, we drop the first term and tune the new leading polynomial $(1 - P_t)^{\gamma+2}$.

Detecting 3D objects from LiDAR point clouds is an important topic and can directly benefit autonomous driving applications. We conduct these experiments on the Waymo Open Dataset (Sun et al., 2020). Similar to 2D detectors, 3D detection models are commonly based on single-stage and two-stage architectures. Here, we evaluate our PolyLoss on two models: a popular single-stage PointPillars model (Lang et al., 2019); and a state-of-the-art two-stage Range Sparse Net (RSN) model (Sun et al., 2021). Both models rely on multi-task loss functions during training. Here, we focus on improving the classification focal loss by replacing it with PolyLoss. Similar to the 2D perception cases, we adopt the Poly-1 formulation to improve upon focal loss, shown in Table 6.

**PolyLoss improves single-stage PointPillars model.** The PointPillars model converts the raw 3D point cloud to a 2D top-down pseudo image, and then detect 3D bounding boxes from the 2D image in a similar way to RetinaNet (Lin et al., 2017). Here, we replace the classification

---

[5]Code at `https://github.com/tensorflow/tpu/tree/master/models/official`

| | Loss | BEV | | 3D | |
|---|---|---|---|---|---|
| | | AP/APH L1 | AP/APH L2 | AP/APH L1 | AP/APH L2 |
| Vehicle (IoU=0.7) | | | | | |
| PointPillars $L_{\text{FL}}$ | $-(1-P_t)^2\log(P_t)$ | 82.5/81.5 | 73.9/72.9 | 63.3/62.7 | 55.2/54.7 |
| PointPillars $L_{\text{Poly-1}}^{FL}$ | $-(1-P_t)^2\log(P_t)-(1-P_t)^3$ | **83.6/82.5** | **74.8/73.7** | **63.7/63.1** | **55.5/55.0** |
| Improvement | - | **+1.1/+1.0** | **+0.9/+0.8** | **+0.4/+0.7** | **+0.3/+0.3** |
| RSN $L_{\text{FL}}$ | $-(1-P_t)^2\log(P_t)$ | 91.3/90.8 | 82.6/**82.2** | 78.4/78.1 | 69.5/69.1 |
| RSN $L_{\text{Poly-1*}}^{FL}$ | $-(1-P_t)^2\log(P_t)-(1-P_t)^3-0.4(1-P_t)^4$ | **91.5/90.9** | **82.7**/82.1 | **78.9/78.4** | **69.9/69.5** |
| Improvement | - | **+0.2/+0.1** | **+0.1**/-0.1 | **+0.5/+0.3** | **+0.4/+0.4** |
| Pedestrian (IoU=0.5) | | | | | |
| PointPillars $L_{\text{FL}}$ | $-(1-P_t)^2\log(P_t)$ | 76.0/62.0 | 67.2/54.6 | 68.9/56.6 | 60.0/49.1 |
| PointPillars $L_{\text{Poly-1}}^{FL}$ | $-(1-P_t)^2\log(P_t)-(1-P_t)^3$ | **77.1/62.9** | **67.7/55.1** | **69.6/57.1** | **60.2/49.3** |
| Improvement | - | **+1.1/+0.9** | **+0.5/+0.5** | **+0.7/+0.5** | **+0.2+0.2** |
| RSN $L_{\text{FL}}$ | $-(1-P_t)^2\log(P_t)$ | 85.0/81.4 | 75.5/72.2 | 79.4/76.2 | 69.9/67.0 |
| RSN $L_{\text{Poly-1*}}^{FL}$ | $-(1-P_t)^2\log(P_t)-(1-P_t)^3+0.2(1-P_t)^4$ | **85.4/81.8** | **75.8/72.5** | **80.2/77.0** | **70.6/67.7** |
| Improvement | - | **+0.4/+0.4** | **+0.3/+0.3** | **+0.8/+0.8** | **+0.7/+0.7** |

Table 7: **PolyLoss improves detection results on Waymo Open Dataset** *validation set*. Two detection models: single-stage PointPillars (Lang et al., 2019) and two-stage SOTA RSN (Sun et al., 2021) are evaluated. Bird's eye view (BEV) and 3D detection average precision (AP) and average precision with heading (APH) at Level 1 (L1) and Level 2 (L2) difficulties are reported. The IoU threshold is set to 0.7 for vehicle detection and 0.5 for pedestrian detection.

focal loss ($\gamma = 2$) with $L_{\text{Poly-1}}^{\text{FL}} = -(1-P_t)^2\log P_t + \epsilon_1(1-P_t)^3$ and adopt the same training schedule optimized for focal loss without any modification[6]. Table 7 shows that $L_{\text{Poly-1}}^{\text{FL}}$ with $\epsilon = -1$ leads to significant improvement on all the metrics for both vehicle and pedestrian models.

**Advancing the state-of-the-art with RSN.** RSN segments foreground points from the 3D point cloud in the first stage, and then applies sparse convolution to predict 3D bounding boxes from the selected foreground points. RSN uses the same focal loss as the PointPillars model, i.e., $L_{\text{FL}} = -(1-P_t)^2\log P_t$. Since the optimal $L_{\text{Poly-1}}^{\text{FL}}$ for PointPillars ($\epsilon_1 = -1$) is equivalent to dropping the first polynomial, we adapt the same loss formulation for RSN and tune the new leading polynomial $(1-P_t)^4$ by defining $L_{\text{Poly-1*}}^{\text{FL}} = -(1-P_t)^2\log(P_t) - (1-P_t)^3 + \epsilon_2(1-P_t)^4$, shown in Figure 7. We follow the same training schedule optimized for focal loss described in Sun et al. (2021) without adjustment. Our results, in Table 7, show that tuning the new leading polynomial improves all metrics (except vehicle detection BEV APH L2) for the SOTA 3D detector.

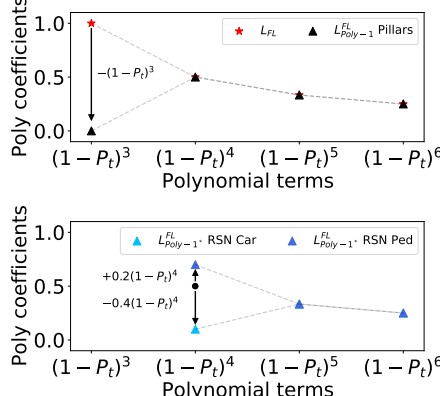

Figure 7: **Visualizing $L_{\text{Poly-1}}^{FL}$ and $L_{\text{Poly-1*}}^{FL}$ in the PolyLoss framework**.

## 6   CONCLUSION

In this paper, we propose the PolyLoss framework, which provides a unified view on common loss functions for classification problems. We recognize that, under polynomial expansion, focal loss is a *horizontal* shift of the polynomial coefficients compared to the cross-entropy loss. This new insight motivates us to explore an alternative dimension. i.e. *vertically* modify the polynomial coefficients.

Our PolyLoss framework provides flexible ways of changing the loss function shape by adjusting the polynomial coefficients. In this framework, we propose a simple and effective *Poly-1* formulation. By simply adjusting the coefficient of the leading polynomial coefficient with just one extra hyperparameter $\epsilon_1$, we show our simple *Poly-1* improves a variety of models across multiple tasks and datasets. We hope Poly-1 formulation's simplicity (one extra line of code) and effectiveness will lead to adoption in more applications of classification than the ones we have managed to explore.

More importantly, our work highlights the limitation of common loss functions, and simple modification could lead to improvements even on well established state-of-the-art models. We hope these findings will encourage exploring and rethinking the loss function design beyond the commonly used cross-entropy and focal loss, as well as the simplest Poly-1 loss proposed in this work.

---

[6]Code at https://github.com/tensorflow/lingvo/tree/master/lingvo/tasks/car

## ACKNOWLEDGEMENTS

We thank James Philbin, Doug Eck, Tsung-Yi Lin and the rest of Waymo Research and Google Brain teams for valuable feedback.

## REPRODUCIBILITY STATEMENT

Our experiments are based on public datasets and open source code repositories, shown in footnote 3-6. We do not tune any default training hyperparameters and only modify the loss functions, which are shown in Table 2-7. The proposed final formulation $L_{\text{Poly-1}}$ requires **one line of code change**. Example code for $L_{\text{Poly-1}}^{\text{CE}}$ with softmax activation is shown below.

```python
def poly1_cross_entropy(logits, labels, epsilon):
    # epsilon >=-1.
    # pt, CE, and Poly1 have shape [batch].
    pt = tf.reduce_sum(labels * tf.nn.softmax(logits), axis=-1)
    CE = tf.nn.softmax_cross_entropy_with_logits(labels, logits)
    Poly1 = CE + epsilon * (1 - pt)
    return Poly1
```

Example code for $L_{\text{Poly-1}}^{\text{CE}}$ with $\alpha$ **label smoothing** is shown below.

```python
def poly1_cross_entropy(logits, labels, epsilon, alpha = 0.1):
    # epsilon >=-1.
    # one_minus_pt, CE, and Poly1 have shape [batch].
    num_classes = labels.get_shape().as_list()[-1]
    smooth_labels = labels * (1-alpha) + alpha/num_classes
    one_minus_pt = tf.reduce_sum(
        smooth_labels * (1 - tf.nn.softmax(logits)), axis=-1)
    CE_loss = tf.keras.losses.CategoricalCrossentropy(
        from_logits=True, label_smoothing=alpha, reduction='none')
    CE = CE_loss(labels, logits)
    Poly1 = CE + epsilon * one_minus_pt
    return Poly1
```

Example code for $L_{\text{Poly-1}}^{\text{FL}}$ with sigmoid activation is shown below.

```python
def poly1_focal_loss(logits, labels, epsilon, gamma=2.0):
    # epsilon >=-1.
    # p, pt, FL, and Poly1 have shape [batch, num of classes].
    p = tf.math.sigmoid(logits)
    pt = labels * p + (1 - labels) * (1 - p)
    FL = focal_loss(pt, gamma)
    Poly1 = FL + epsilon * tf.math.pow(1 - pt, gamma + 1)
    return Poly1
```

Example code for $L_{\text{Poly-1}}^{\text{FL}}$ with $\alpha$ **balance** is shown below.

```python
def poly1_focal_loss(logits, labels, epsilon, gamma=2.0, alpha=0.25):
    # epsilon >=-1.
    # p, pt, FL, weight, and Poly1 have shape [batch, num of classes].
    p = tf.math.sigmoid(logits)
    pt = labels * p + (1 - labels) * (1 - p)
    FL = focal_loss(pt, gamma, alpha)
    weight = labels * alpha + (1 - labels) * (1 - alpha)
    Poly1 = FL + epsilon * tf.math.pow(1 - pt, gamma + 1) * weight
    return Poly1
```

SUPPLEMENTARY MATERIAL

## 7 PROOF OF THEOREM 1

**Theorem 1.** *For any small $\zeta > 0$, $\delta > 0$ if $N > \log_{1-\delta}(\zeta \cdot \delta)$, then for any $p \in [\delta, 1]$, we have $|R_N(p)| < \zeta$ and $|R'_N(p)| < \zeta$.*

*Proof.*

$$|R_N(p)| = \sum_{j=N+1}^{\infty} 1/j(1-p)^j \leq \sum_{j=N+1}^{\infty} (1-p)^j = \frac{(1-p)^{N+1}}{p} \leq \frac{(1-\delta)^{N+1}}{\delta} \leq \frac{(1-\delta)^N}{\delta}$$

$$|R'_N(p)| = \sum_{j=N}^{\infty} (1-p)^j = \frac{(1-p)^N}{p} \leq \frac{(1-\delta)^N}{\delta}$$

## 8 ADJUSTING OTHER TRAINING HYPERPARAMETERS LEADS TO HIGHER GAIN.

All the experiments shown in the main text are based on hyperparameters optimized for the baseline loss function, which actually puts PolyLoss at a disadvantage. Here we use weight decay rate for ResNet50 as an example. The default weight decay (1e-4) is optimized for cross-entropy loss. Adjusting the decay rate may reduce the model performance of cross-entropy loss but leads to much higher gain for PolyLoss (+0.8%), which is better than the best accuracy (76.3%) trained using cross-entropy loss (+0.8%).

| Weight decay | 1e-4[†] | 2e-4 | 9e-5 |
|---|---|---|---|
| Cross-entropy | **76.3** | **76.3** | 76.1 |
| PolyLoss | 76.7 | **77.1** | 76.7 |
| Improv. @ the same weight decay | +0.4 | **+0.8** | +0.6 |
| Improv. compared to the best $L_{\text{CE}}$ (76.3%) | +0.4 | **+0.8** | +0.4 |

Table 8: **ResNet50 performances on ImageNet-1K using different weight decays.** [†]The default weight decay value is 1e-4.

Here, we add additional ablation studies on COCO detection using RetinaNet. The optimal $\gamma$ and $\alpha$ balance values for Focal loss are (2.0, 0.25) (Lin et al., 2017). Since all the hyperparameters are optimized with respect to the optimal $(\gamma, \alpha)$ values, we observe no improvement when tuning the leading polynomial term. We suspect the detection AP is at a 'local maximum' of hyperparameters. By adjusting $(\gamma, \alpha)$ values, we show PolyLoss consistently outperforms the best Focal Loss AP (33.4), i.e., adjusting only $\gamma$ value (column 3, 4) or both $\gamma$ and $\alpha$ values (column 5, 6).

| Focal loss $(\gamma, \alpha)$ | $(2.0, 0.25)^{†}$ | $(1.5, 0.25)$ | $(2.5, 0.25)$ | $(1.5, 0.3)$ | $(2.5, 0.15)$ |
|---|---|---|---|---|---|
| Focal loss | **33.4** | **33.4** | 33.2 | 33.2 | 32.9 |
| PolyLoss | 33.4 | 33.6 | 33.7 | **33.8** | **33.8** |
| Improv. @ same $(\gamma, \alpha)$ | 0 | +0.2 | +0.5 | +0.6 | +0.9 |
| Improv. compared to the best $L_{\text{FL}}$ (33.4) | 0 | +0.2 | +0.3 | **+0.4** | **+0.4** |

Table 9: **RetinaNet (ResNet50 backbone) performances on COCO using different Focal loss** $(\gamma, \alpha)$. [†]The default $(\gamma, \alpha)$ used in Focal loss is (2.0, 0.25).

## 9 $L_{\text{DROP}}$ WITH MORE HYPERPARAMETER TUNING

For $L_{\text{Drop}}$ (N = 2), besides adjusting the learning rate, we further tune the coefficient ($\alpha$) of the second polynomial, similar to a prior work (Gonzalez & Miikkulainen, 2020b), and weight decay.

$$L_{\text{Drop*}} = (1 - P_t) + \alpha(1 - P_t)^2 \tag{8}$$

Unlike Feng et al. (2020), where $\alpha = 0.5$ after dropping all higher-order polynomial, we find the optimal $\alpha = 8$, while the optimal learning rate is the same as the default setting (0.1). This alone

increases the accuracy to 70.9, which shows simply dropping polynomial terms is not enough and adjusting the polynomial coefficients is critical. Further tuning weight decay leads to less than 0.1% model quality improvement.

Comparing to prior works (Gonzalez & Miikkulainen, 2020b; Feng et al., 2020), Poly-1 is more effective and only contains one hyperparameter. Tuning weight decay of Poly-1 further increases the accuracy while having less hyperparameters compared to $L_{\mathrm{Drop}^*}$, shown in Table 10.

|  | Cross-entropy | Poly-1 | Poly-1 (weight decay) | $L_{\mathrm{Drop}^*}$ |
|---|---|---|---|---|
| Accuracy | 76.3 | 76.7 | **77.1** | 70.9 |
| Num. of parameters | – | 1 | 2 | 3 |

Table 10: **Poly-1 outperforms $L_{\mathbf{Drop}^*}$ with hyperparameter tuning.** Accuracy of ResNet50 on ImageNet-1K is reported.

## 10 COLLECTIVELY TUNING MULTIPLE POLYNOMIAL COEFFICIENTS

Besides adjusting individual polynomial coefficients, in this section, we explore collectively tuning multiple polynomial coefficients in the PolyLoss framwork. In particular, we change the coefficients in the original cross-entropy loss from $1/j$ (Equation 1) to exponential decay. Here, we define

$$L_{\mathrm{exp}} = \sum_{j=1}^{2N} e^{-(j-1)/N}(1 - P_t)^j \tag{9}$$

where we cut off the infinite sum at twice the decay factor $N$. We performed 2D grid search on $N \in \{5, 20, 80, 320\}$ and learning rate $\in \{0.1, 0.4, 1.6, 6.4\}$. The best accuracy is 72.3, where $N = 80$ and learning rate $= 1.6$, shown in Table 11.

|  | Cross-entropy | Poly-1 | $L_{\mathrm{exp}}$ |
|---|---|---|---|
| Accuracy | 76.3 | **76.7** | 72.3 |
| Num. of parameters | – | 1 | 2 |

Table 11: **Comparing Poly-1 with exponential decay coefficients.** Accuracy of ResNet50 on ImageNet-1K is reported.

Though Poly-1 is better than using $L_{\mathrm{exp}}$, there are a lot more possibilities besides using exponential decay. We believe understanding how collectively tuning multiple coefficients affects the training is an important topic.

## 11 COMPARING TO OTHER TRAINING TECHNIQUES

As shown in recent works (He et al., 2019; Bello et al., 2021; Wightman et al., 2021), though independent novel training techniques often lead to sub 1% improvement, combining them could lead to significant overall improvements. To put things into perspective, Poly-1 achieves similar improvements as other commonly used training techniques, such as label smoothing and dropout on FC, shown in Table 12.

|  | Cross-entropy | Poly-1 | Label smoothing | Dropout on FC |
|---|---|---|---|---|
| Accuracy | 76.3 | **76.7** | 76.7 | 76.4 |
| Num. of parameters | – | 1 | 1 | 1 |

Table 12: **Comparing Poly-1 with common training techniques.** Accuracy of ResNet50 on ImageNet-1K is reported.

## 12 Rediscovering focal loss from PolyLoss

Focal loss was first developed for single-stage detector RetinaNet to address strong class imbalance presented in object detection (Lin et al., 2017). Here, we provide an additional ablation study on how to systemically discover focal loss in the PolyLoss framework and investigate how the leading terms affect training in the presence of class imbalance.

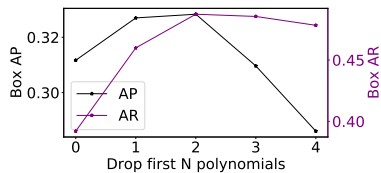

Figure 8: **Dropping leading polynomial terms can improve RetinaNet.**

**Rediscovering the concept of focal loss from cross-entropy loss.** Here, we take a step back and attempt to systematically rediscover the concept of focal loss via our PolyLoss framework. Focal loss is commonly used for training detection models. Coming up with such an insight to address the class imbalance issue in detection requires strong domain expertise. We start with the PolyLoss representation of cross-entropy loss and improve it from the PolyLoss gradient perspective.

We start with the cross-entropy loss and define PolyLoss by dropping the first $N$ polynomials in cross-entropy loss, i.e. $L_{\text{Drop-front}} = \sum_{j=N+1}^{\infty} 1/j(1 - P_t)^j = L_{\text{CE}} - \sum_{j=1}^{N} 1/j(1 - P_t)^j$. Dropping the first two polynomial terms $(1 - P_t)$ significantly improves both the detection AP and AR, see Figure 8. Dropping the first two polynomials ($N = 2$) leads to the best RetinaNet performance, which is similar to setting $\gamma = 2$ in focal loss, i.e. focal loss $\gamma = 2$ pushes all the polynomial coefficients to the right by 2, shown in Figure 1 right, which is similar to truncating the first two polynomial terms.

**Leading polynomials cause overfitting to the majority class.** In the PolyLoss framework, the leading polynomial of cross-entropy loss is a constant, shown in Equation 3. For binary classification, the leading gradient for each class is simply $N_{background} - N_{object}$, where $N_{background}$ and $N_{object}$ are the counts of background and object instances in the training mini-batch. When the class counts are extremely imbalanced, the majority class will dominate the gradient which will lead to significant bias towards optimizing the majority class.

Dropping polynomials reduces the extremely confident prediction $P_t$, see Figure 9. To examine the composition of the overall prediction confidence, we also plot the $P_t$ for background only and $P_t$ for object only. Due to the extreme imbalance between the background and

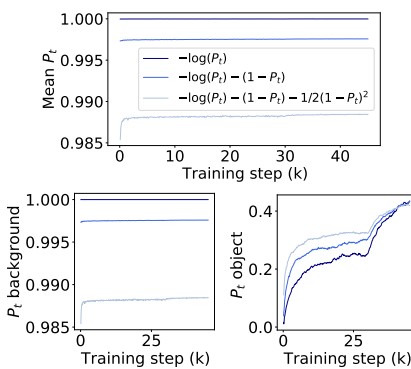

Figure 9: **Dropping leading polynomials reduces overfitting to the majority class.** $P_t$ during RetinaNet training are plotted. Top: overall. Bottom left: background. Bottom right: foreground object. Dark blue curves represents $P_t$ for cross-entropy loss. Blue curves represents dropping the first polynomial in the cross-entropy loss. Light blue curves represents dropping both the first and second polynomials in the cross-entropy loss.

the object class, the overall $P_t$ is dominated by the background only $P_t$. So reducing the overall $P_t$ decreases the background $P_t$. On the other hand, reducing overfitting to the majority background class leads to more confident prediction $P_t$ on the object class.

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
