# OpenReview forum: "PolyLoss: A Polynomial Expansion Perspective of Classification Loss Functions"
_ICLR.cc/2022/Conference — ICLR 2022 Poster_

### Official Review · Reviewer_Vgzk · 2021-10-26

**Correctness:** 3
**Technical Novelty And Significance:** 3
**Empirical Novelty And Significance:** 3
**Recommendation:** 6
**Confidence:** 4

**Main Review:**

Strengths:
- Clear and easy to follow text.
- An application of the Taylor expansion of cross entropy loss and focal loss on different classification problems.
- Experiments on different classification problems.

Weaknesses:
- The major problem that I had with the first version, which persists with the current version, is that some results are not significant. The authors obtain very minor improvements on several models and tasks. The large values for AR are actually not impressive since many simple tricks can provide similar (even better) improvements for AR. And the reported improvements for all other measurements and tasks are rather minimal.

Minor comment:
- "can improves" => "can improve".

**Summary Of The Paper:**


In this paper, the authors introduce a new loss function for classification problems. To be specific, the authors introduce Taylor expansion of cross-entropy loss + focal loss and show that various subsets of this expansion can improve the models on image classification, 2D object detection and 3D object detection problems.

I've reviewed a previous version of this paper at NeurIPS2021. Compared to the previous version, I see that the authors improved the paper significantly, mainly in terms of structure and presentation.

**Summary Of The Review:**

In general, I am positive about the paper, though I need to be convinced about the significance of the results and the importance of the work.

---

> ### Author Response · Authors · 2021-11-12
> **Response to Reviewer Vgzk**
>
> We sincerely appreciate your thoughtful feedback from the previous conference as well as the current one. We are glad you acknowledge the improvements. We hope the replies below address your concerns and will modify the paper accordingly. Please let us know if you have any other comments. Thank you!
>
> > Significance of the results and the importance of the work
>
> The main motivation for this paper is not to demonstrate Poly-1 leads to significant absolute improvements in all the tasks. Instead, we would like to introduce the PolyLoss framework and raise the awareness of the limitation of common loss functions (Figure 1). Poly-1 is the simplest possible exploration of vertically adjusting the polynomial coefficient, and still improves upon the common loss functions in all the tasks we tried.
>
> We will clarify in the paper that Poly-1 is not the destination of PolyLoss but a starting point for designing loss function from the prospective of polynomial expansion. Besides showing Poly-1 can improve model performance, we provide new insights and findings on why the leading polynomial term is important (Figure 3) and the connection between the leading term and the prediction confidence (Figure  5 and 6). There are also potentially theoretical questions posed by this work --- when do we need positive or negative scaling of the first polynomial, and whether this is potentially tied to model score miscalibration during training. This could be further explored in future work --- e.g. keep tracking how calibrated the scores are during training on a validation/testing set as the training progresses and cross-correlate that to method gains. We hope the results and findings in this work could encourage more exploration on loss function design beyond the commonly used cross-entropy and focal loss, as well as the Poly-1 loss proposed in this work.
>
> To put things into perspective, Poly-1 achieves similar improvements as other commonly used training techniques, such as label smoothing and dropout on FC.
>
> |  | Cross-entropy | Poly-1 |Label smoothing | Dropout on FC|
> |---|---|---|---|---|
> | Accuracy | 76.3 | 76.7 (+0.4) | 76.7 (+0.4) | 76.4 (+0.1)|
> | # of parameters |  | 1 | 1 | 1  | 1|
>
> RTable 1. Comparing Poly-1 with common training techniques on ImageNet1K using ResNet-50.
>
> As shown in recent works [1-3], though independent novel training techniques often lead to sub 1% improvement, combining them could lead to significant overall improvements. We hope, given its simplicity and generalizability, Poly-1 could become a new training ingredient for improving model performance just like label smoothing, dropout, etc.
>
> > "can improves" => "can improve".
>
> Thank you for pointing it out. We will fix it in the paper.
>
> [1] He, Tong, et al. "Bag of tricks for image classification with convolutional neural networks." Proceedings of the IEEE/CVF Conference on Computer Vision and Pattern Recognition. 2019.
>
> [2] Bello, Irwan, et al. "Revisiting resnets: Improved training and scaling strategies." arXiv preprint arXiv:2103.07579 (2021).
>
> [3] Wightman, Ross, Hugo Touvron, and Hervé Jégou. "ResNet strikes back: An improved training procedure in timm." arXiv preprint arXiv:2110.00476 (2021).

---

### Official Review · Reviewer_Zps3 · 2021-11-01

**Correctness:** 3
**Technical Novelty And Significance:** 2
**Empirical Novelty And Significance:** 2
**Recommendation:** 6
**Confidence:** 2

**Main Review:**

# Strengths

The paper reads well and has almost no typos. It provides an interesting observation showing how different losses could all be expressed in a common polynomial form where only coefficients differ.

# Weaknesses

The motivation is not entirely clear. While loss functions could be expressed in the form of Taylor expansions, the fact that the polynomial coefficients would have to be fine-tuned in order to obtain further performance improvements introduces even more hyper-parameters for in model giving even more room for possible overfitting. Moreover, the claimed improvements are only marginally higher than the chosen baselines. It is not entirely clear if the added complexity (in both having to fine-tune the coefficients as well as determining where the expansions should be cut) would be justifiable.

## Minor

 - "polynmoial" (p.9, conclusion)

**Summary Of The Paper:**

The paper proposes a framework for creating loss functions based on the polynomial expansion of known loss functions. The authors show that, by fine-tuning the polynomial coefficients of those expansions can bring improvements in multiple computer vision tasks. The authors experiment with image classification (ImageNet-21k), instance segmentation and object detection (COCO), 3D object detection (Waymo), showing small improvements over the chosen baselines.



**Summary Of The Review:**

The paper is interesting, but it is unclear to me whether replacing losses with their polynomial expansions while bringing only small performance improvements to existing methods would suffice for a publication in ICLR.

After rebuttal, most of my concerns have been addressed. I am therefore increasing my rating to 6.

---

> ### Author Response · Authors · 2021-11-12
> **Response to Reviewer Zps3 [Part 1]**
>
> We would like to thank you for the valuable comments and are glad that you find our work interesting. We hope the replies below will resolve your concerns. Please let us know if you have any other comments. Thank you!
>
> > Q1: The motivation is not entirely clear.
>
> Cross-entropy loss and focal loss are widely used for training deep learning models, but there is not much mathematical understanding and exploration for such losses: what are the connections between them? And are they already optimal? Especially for focal loss, gains are inconsistent on different datasets. For example, on the **imbalanced** ImageNet-21K, we find focal loss is suboptimal compared to cross entropy loss.
>
> |  | Cross-entropy | Poly-1 | Focal|
> |---|---|---|---|
> | Accuracy | 43.5 | **44.5**  | 43.4 |
>
> RTable1: Comparing cross-entropy loss, Poly-1 loss and focal loss on ImageNet21K using EffNetV2-S [3].
>
> In this paper, we try to question this common belief. We start with decomposing cross-entropy loss and focal loss in the PolyLoss framework so that we can understand the connection between them and improve them.
>
> Our studies show that the leading polynomial contributes more than half the gradient in 65% of the training steps (Figure 3 (b)) and its importance also varies for different applications (classification, 2D detection, 2D segmentation and 3D detection). These motivate us to propose the final Poly-1 loss formulation that can be adapted to different applications with only one hyperparameter.
>
> Besides showing Poly-1 is simple and effective, we hope to raise the awareness of the limitations of common loss functions (Figure 1). More importantly, we hope our demonstrations provide encouragement for the machine learning community to continue to explore and rethink the loss function design beyond the commonly used cross-entropy and focal loss, as well as the Poly-1 loss proposed in this work.
>
> >Q2: While loss functions could be expressed in the form of Taylor expansions, the fact that the polynomial coefficients would have to be fine-tuned in order to obtain further performance improvements introduces even more hyper-parameters for in model giving even more room for possible overfitting.
>
> We agree that simplicity is a merit, and that’s why we focus on the simplest form, Poly-1, which only has ONE constant hyperparameter $\epsilon_1$.  Unlike other training hyperparameters like learning rate, our  $\epsilon_1$ is a constant value for all training steps, and thus much easier to tune.
>
> >Q3: Moreover, the claimed improvements are only marginally higher than the chosen baselines. It is not entirely clear if the added complexity (in both having to fine-tune the coefficients as well as determining where the expansions should be cut) would be justifiable.
>
> We would like to emphasize that our **Poly-1 only contains ONE constant hyperparameter** $\epsilon_1$ (Table 2 and Eqn. 7), and it doesn’t need to determine where the expansion should be cut. Previous works [1] [2] require tuning multiple polynomials or expansion cutoff.
>
> We argue that our simple **Poly-1 generalizes to different models and tasks** such as image classification, 2D detection, and 3D detection, sometimes shows  large improvements. For example, in Figure 4, our **Poly-1 improves the well-established SOTA EffNetV2 [3] significantly, which is equivalent to reducing the computational cost by 2.2x.** In particular, the same Poly-1 $L_{Poly-1} = -\log(P_t) + 5(1-P_t)$ improves all three EffNetV2-{S, M, L} models, which shows Poly-1 can generalize to different models (Figure 4).
>
> That being said, in the main text, we highlight that **we do not need to tune other training hyperparameters when using Poly-1**, including very well tuned state-of-the-art models.  Since other training hyperparameters are optimized for cross-entropy loss or focal loss, Poly-1 is disadvantaged. As shown in the appendix B Table 8,  adjusting training parameters such as weight decay doubles improvement from the Poly-1.
>
> To put things into perspective, **Poly-1 achieves similar improvements as other commonly used training techniques**, such as label smoothing and dropout on FC.
>
> |  | Cross-entropy | Poly-1 |Label smoothing | Dropout on FC|
> |---|---|---|---|---|
> | Accuracy | 76.3 | 76.7 (+0.4) | 76.7 (+0.4) | 76.4 (+0.1)|
> | # of parameters |  | 1 | 1 | 1  | 1|
>
> RTable 2. Comparing Poly-1 with common training techniques on ImageNet1K using ResNet-50.
>
> As shown in recent works [4-6], though independent novel training techniques often lead to sub 1% improvement, combining them could lead to significant overall improvements. We hope, given its simplicity and generalizability, Poly-1 could become a new training ingredient for improving model performance just like label smoothing, dropout, etc.
>
> > "polynmoial" (p.9, conclusion)
>
> Thank you for pointing it out. We will fix it in the paper.

---

> > ### Author Response · Authors · 2021-11-12
> > **Response to Reviewer Zps3 [Part 2]**
> >
> > [1] Feng, Lei, et al. "Can cross entropy loss be robust to label noise?." IJCAI. 2020.
> >
> > [2] Gonzalez, Santiago, and Risto Miikkulainen. "Evolving loss functions with multivariate taylor polynomial parameterizations." arXiv preprint arXiv:2002.00059 (2020).
> >
> > [3] Tan, Mingxing, and Quoc V. Le. "Efficientnetv2: Smaller models and faster training." arXiv preprint arXiv:2104.00298 (2021).
> >
> > [4] He, Tong, et al. "Bag of tricks for image classification with convolutional neural networks." Proceedings of the IEEE/CVF Conference on Computer Vision and Pattern Recognition. 2019.
> >
> > [5] Bello, Irwan, et al. "Revisiting resnets: Improved training and scaling strategies." arXiv preprint arXiv:2103.07579 (2021).
> >
> > [6] Wightman, Ross, Hugo Touvron, and Hervé Jégou. "ResNet strikes back: An improved training procedure in timm." arXiv preprint arXiv:2110.00476 (2021).

---

### Official Review · Reviewer_kfqh · 2021-11-03

**Correctness:** 4
**Technical Novelty And Significance:** 3
**Empirical Novelty And Significance:** 3
**Recommendation:** 8
**Confidence:** 5

**Main Review:**

- (a) Paper is well-written, especially the main part of PolyLoss, which has a clear logic.
- (b) The paper has an interesting idea that shows cross-entropy and focal loss can be expressed as a linear combination of polynomial functions. Then it unifies these two loss functions as a part of the general polynomial form. The final PolyLoss is simple and effective.
- (c) The experiments are solid enough to support the main idea.

**Summary Of The Paper:**

- **Motivation**.
The paper argues that the original cross-entropy loss and focal loss are primarily used in the classification task.
A general loss function should get rid of constraints of learning tasks and datasets.


- **Method**. Motivated by this insight, the paper expresses the loss function as a linear combination of polynomial functions and shows that cross-entropy loss and focal loss are special cases.
Then inspired by Taylor expansion, the paper proposes a polynomial loss function, called PolyLoss.
The final version $\text{Poly}-1$ is with first-order.


- **Experiments**. The paper verifies the proposed loss on ImageNet for classification, MS-COCO for detection and instance segmentation, and Waymo Open dataset for 3D detection.

**Summary Of The Review:**

I like this paper. The idea is new, and the discussion is deep. I have no comments on how to improve the paper. Although partial experimental improvements are marginal, I think the novelty is promised. It deserves to expose to the community to inspire future works.

---

> ### Author Response · Authors · 2021-11-12
> **Response to Reviewer kfqh**
>
> We would like to thank you for taking the time to review our paper! We are very grateful for your positive comments acknowledging the novelty as well as the constructive feedback.
>
> Besides showing Poly-1 is simple and effective, we hope to raise the awareness of the limitations of common loss functions (Figure 1).  More importantly, we hope our demonstrations provide encouragement for the machine learning community to continue to explore and rethink the loss function design beyond the commonly used cross-entropy and focal loss, as well as the Poly-1 loss proposed in this work.
>
> Please let us know if you have any other comments/questions. Thank you!

---

### Official Review · Reviewer_KpJa · 2021-11-07

**Correctness:** 4
**Technical Novelty And Significance:** 3
**Empirical Novelty And Significance:** 2
**Recommendation:** 6
**Confidence:** 3

**Main Review:**

## Strengths
(1) The polynomial expansion perspective is interesting and the PloyLoss makes sense.

(2) Extensive experiments are provided and further produce some findings.

(3) The proposed loss function can improve various tasks: image classification, instance segmentation, and 3D object detection.

## Weaknesses
(1) surprisingly, the final version of PloyLoss is Ploy-1. I think this final version is trivial. Does this mean the high-order items are useless? I understand the results of $L_{Drop}$ in Figure 2(a), but I still think the poor results of the low-order loss function are because of the bad coefficient tuning. So can you provide the following versions for comparison?
- L1 loss with hyper-parameter tuning. (Drop the CE part in the Ploy-1.)
- L1 and L2 loss with the hyper-parameter tuning.

(2) Because this paper proposes a general loss family, the exploration is still insufficient. There are many variables of the PloyLoss family, such as the coefficient (particularly analysis in this paper), the start order ($\gamma$ in Focal loss), and the coefficient decay strategy.
- CE and focal loss only provides one kind of coefficient organization strategy as the logarithmic function. What about others? For example, the exponential decay for the coefficients.

(3) The name is misleading for $L_{ploy-1}$ in image classification and $L_{ploy-1}$ in object detection since they use different base loss functions (CE and focal respectively).

(4) It seems like some errors in the label of the y-axis in Figure 2(b).

**Summary Of The Paper:**

This paper analyzes the cross-entropy loss and focal loss through the polynomial expansion perspective. Further, this paper proposes a family of loss functions called PloyLoss. Three instances of PloyLoss are analyzed and obtain the final version as ploy-1. Detailed and extensive experiments are provided.

**Summary Of The Review:**

This paper starts from an interesting loss family as PloyLoss, but the final version is quite simple and trivial. Considering both the experiments, motivation, and the final method, I would like to rank this paper as 5. I am positive about this paper, but more experiments and clarification are expected.

After rebuttal, most of my concerns are addressed. Thus, I raise the rank to 6.

---

> ### Author Response · Authors · 2021-11-12
> **Response to Reviewer KpJa [Part 1]**
>
> We would like to thank you for the thoughtful review and are glad that you find our work interesting. We address the questions you have raised in detail below.  Please let us know if you have any other comments. Thank you!
>
> > Q1. Surprisingly, the final version of PloyLoss is Ploy-1. I think this final version is trivial.
>
> We intend to focus on Poly-1 as it is the simplest form of PloyLoss (only one hyperparameter $\epsilon_1$) and to raise the awareness of the limitations of common loss functions (Figure 1).  Although Poly-1 is simple, it is not straightforward to arrive at this simple solution without the PolyLoss framework.  It is possible only if we expand the loss with polynomial terms and identify the importance of the leading term in gradient update, shown in Figure 3 (b).
>
> Notably, our PloyLoss framework is flexible to allow users to choose more complex forms: for example, Poly-2 achieves even better performance (Table 3), at the cost of more hyperparameters ($\epsilon_2$).  We believe tuning more terms with better tuning algorithms beyond grid search will lead to even higher gain.
>
> More importantly, we hope our demonstrations provide encouragement for the machine learning community to continue to explore and rethink the loss function design beyond the commonly used cross-entropy and focal loss, as well as the Poly-1 loss proposed in this work.
>
> > Q2: Does this mean the high-order items are useless?
>
> Higher order items are important as they contribute more than 50% of the gradient in the early training steps (Theorem 1 and Figure3). Previous approaches simply drop the high-order items and lead to significant accuracy loss (Figure 2).
> Our Poly-1 loss keeps all higher order items (Table 2).
>
> If we aim to improve common loss functions with minimal tuning, which polynomial coefficient should we tune? Our study (Figure 3 (b)) shows the leading polynomial term contributes to more than 50% of  the gradient at the last 65% of training steps. That's why we focus on tuning the first polynomial coefficient, instead of tuning higher-order terms. We also attempt to tune coefficients for higher-order items, but it introduces more hyperparameters and has diminished gain.
>
> Again, PolyLoss framework allows users to design different forms of loss, and this paper focuses on the simplest Poly-1 to raise the awareness of the limitations of common loss functions (Figure 1).
>
> >Q3: I understand the results of  LDrop  in Figure 2(a), but I still think the poor results of the low-order loss function are because of the bad coefficient tuning. So can you provide the following versions for comparison? (1) L1 loss with hyper-parameter tuning. (Drop the CE part in the Ploy-1.) (2) L1 and L2 loss with the hyper-parameter tuning.
>
> Yes, we agree. LDrop section is to show simply dropping high-order polynomial terms might not be the best way to design the loss function, and potentially requires careful tuning of multiple parameters. We follow the same implementation as in [1], which has two parameters: polynomial cutoff thresholds and learning rate.
>
> As requested by you, we perform more hyperparameter tuning for (1) L1, (2) L1 and L2 loss, which are summarized below
>
> * Larger learning rate search space $\in$ [0.005,0.01, 0.05, 0.1, 0.2, 1, 5, 10, 20]
> * Weight decay $\in$ [0.1, 0.3, 0.5, 0.7, 0.9, 1.2, 2, 3] $\times$ 1e-4
>
> In addition, for L1 and L2 loss, we also
>
> * Tune the ratio between L1 and L2, i.e. loss = L1 + $\alpha$ L2, where $\alpha$ $\in$ [0, 0.1, 0.2, 0.4, 0.8, 1.0, 1.2, 2, 4, 8, 10]
>
> For L1 loss with hyperparameter tuning, the best we get is 1.3%.
>
> For L1 and L2 loss with hyperparameter tuning, unlike [1], where $\alpha = 0.5$ after dropping all higher-order polynomial, we find the optimal $ \alpha = 8$, while the optimal learning rate is the same as the default setting (LR=0.1). This alone increases the accuracy to 70.9, which shows simply dropping polynomial terms is not enough and adjusting the polynomial coefficients is critical. Tuning weight decay leads to less than 0.1% model quality improvement.
>
> |  | Cross-entropy | Poly-1 | Poly-1 (weight decay) | L1 | L1 and L2 |
> |---|---|---|---|---|---|
> | Accuracy | 76.3 | 76.7 | 77.1 | 1.3 | 70.9 |
> | # of parameters | -- | 1 | 2 | 2 | 3 |
>
> RTable1: Comparing L1 loss, L1 and L2 loss with Poly-1 and cross-entropy loss on ImageNet1K using ResNet-50. More details on Poly-1 (weight decay) are shown in Appendix B Table 8.
>
> As shown in RTable1, Poly-1 requires minimal tuning while being more effective than L1 loss, L1 and L2 loss.

---

> > ### Author Response · Authors · 2021-11-12
> > **Response to Reviewer KpJa [Part 2]**
> >
> > > Q4: Because this paper proposes a general loss family, the exploration is still insufficient. There are many variables of the PloyLoss family, such as the coefficient (particularly analysis in this paper), the start order (γ  in Focal loss), and the coefficient decay strategy.
> > CE and focal loss only provides one kind of coefficient organization strategy as the logarithmic function. What about others? For example, the exponential decay for the coefficients.
> >
> > We are very glad you ask this question. Besides Poly-1, we hope the PolyLoss framework will encourage researchers to rethink loss function design from the perspective of polynomial coefficients. As you point out, besides decaying coefficients $(1/j)$  in cross-entropy loss, there are other strategies such as exponential decay ($e^{-j}$).
> >
> > Here, we define loss as $ L = \sum_{j=1}^{2N} e^{-(j-1)/N} \times (1 -P_t)^j$, where we cut off the infinite sum at twice the decay factor N.  We performed 2D grid search on
> > * N $\in$ [5, 20, 80, 320]
> > * Learning rate $\in$ [0.1, 0.4, 1.6, 6.4].
> >
> > The best accuracy is 72.3, where N = 80 and LR = 1.6, shown in RTable 2.
> >
> > |  | Cross-entropy | Poly-1 | Exponential Decay |
> > |---|---|---|---|
> > | Accuracy | 76.3 | 76.7 | 72.3  |
> > | # of parameters | -- | 1 | 2 |
> >
> > RTable 2: Comparing different strategies of coefficient organization on ImageNet1K using ResNet-50.
> >
> >
> > Though Poly-1 is better than using exponential decay coefficients, we believe understanding how collectively tuning multiple coefficients affects the training is an important topic. There are a lot more possibilities besides using exponential decay, which we plan to explore in future works.
> >
> > >Q5: The name is misleading for  Lploy−1 in image classification and  Lploy−1  in object detection since they use different base loss functions (CE and focal respectively).
> >
> > Thank you for pointing it out. We will change Poly-1 for focal loss to $L_{Poly-1}^{FL}$
> >
> > > Q6: It seems like some errors in the label of the y-axis in Figure 2(b).
> >
> > Those are correct. The quality of the model suffers a lot if we simply truncate higher-order terms.
> >
> > [1] Feng, Lei, et al. "Can cross entropy loss be robust to label noise?." IJCAI. 2020.

---

> > > ### Comment · Reviewer_KpJa · 2021-11-14
> > > **Thanks for the response**
> > >
> > > I would like to thank the authors for responding to my previous questions. The results of Q3 and Q4 are convincing. Most of my concerns are addressed.
> > >
> > > Since this website allows the authors to upload a new version of the paper, I suggest the author do it. It will be more clear to see the revised paper.

---

> > > > ### Author Response · Authors · 2021-11-15
> > > > **Reversion uploaded.**
> > > >
> > > > We would like to thank you for the prompt reply and increasing the score! We have uploaded a new version of the paper.
> > > >
> > > > We added additional ablation studies proposed by you in appendix C (for $L_{Drop}$) and D (for exponential decay coefficients).  We also changed Poly-1 for focal loss to $L_{Poly-1}^{FL}$ We will send out a global summary once we hear back from all reviewers or before the rebuttal deadline.
> > > >
> > > > Please let us know if you have any other comments/suggestions.

---

### Author Response · Authors · 2021-11-23
**To all reviewers**

We thank all reviewers for their thoughtful comments and feedbacks. We are excited that all reviewers see the paper favorably. In the rebuttal revision, the major text changes are marked in blue, including:

1. Emphasizing Poly-1 is the simplest loss formulation in the PolyLoss framework in **Table 1** (Reviewer Zps3)
2. Clarifying the motivation/importance of this work in **Conclusion**  (Reviewers Zps3 and Vgzk)
3. $L_{Drop}$ with more hyperparameter tuning in **Appendix C** (Reviewer KpJa)
4. Collectively tuning multiple coefficients in **Appendix D** (Reviewer KpJa)
5. Clarifying the significance of Poly-1 is on par with other common training techniques in **Appendix E** (Reviewers Zps3 and Vgzk)

---

### Public Comment · ~Guangxiang_Zhao3 · 2022-04-12
**The additional polynomial loss (e.g. 1-p)  function has been proposed in a previous paper.**

Hi there. Congratulations on your paper being accepted. We appreciate your new explanation of polynomial loss.

**However, the additional polynomial loss (e.g., $1-p$)  function has been proposed in our previous paper [1.2].  And we provide more effective variants and provide theoretical explanations.  But we are not cited and discussed in this paper.**

We have a paper that was online in the ICLR 2021 submission [1] that proposes the polynomial form of $p$  as an additional term（$-p^\gamma$）. We name the additional polynomial  loss as Power Bonus (see Section 5.3 and Figure 3 in paper [1]),

This idea was finally published in AAAI 2022 [2].  We propose Encouraging Loss: $-logp+log(1-p)$ and  design many types of the additional terms that approximate $log(1-p)$ by **E**nd the **L**og curve in the high-likelihood region and replace
the log curve with the tangent of the endpoint.



\begin{equation}
f(p)=-log p+ log (1-p) \quad if \quad p \leq LE \quad or \quad -log p - \frac{p-LE}{1-LE} +log(1-LE) \quad if \quad  p > LE.
\end{equation}


**The best choice in this paper (i.e. $1-p$) is a moderate choice in our paper**， that is Encouraging Loss with LE=0, i.e. $-logp-p$, **our other variants get better performance (See Figure 4 in paper [2]).**

**More than the exclusive results, our explanation challenges unconventional wisdom and finds this idea meets desired theoretical properties.** We find that well-classified examples are underestimated in classification and the optimization of these examples can improve representation learning, energy optimization, and margin.

**In our explanation, the additional loss term does not need to be polynomial.** As long as it is convex to restore the learning of well-classified samples and not generate excessive gradients. (e.g., Encouraging Loss with LE=0.5)

In the high-performance area, Encouraging Loss improves OFA with the base arch of Vit-Large by 0.6 (85.0->85.6) on training ImageNet-1K from scratch and VQA (80.1->80.7). Refer to https://github.com/OFA-Sys/OFA.


[1] High-Likelihood Area Matters --- Rewarding Correct, Rare Predictions Under Imbalanced Distributions.  ICLR 2021 submission. https://openreview.net/pdf?id=7Yhok3vJpU

[2] Well-classified Examples are Underestimated in Classification with Deep Neural Networks.  AAAI 2022. https://arxiv.org/abs/2110.06537

Minor: We can also multiply the whole loss or the additional term with class weights to adjust it for class imbalanced learning.

---

> ### Public Comment · ~Zhaoqi_Leng1 · 2022-04-12
> **Adjusting (1-P)^j is discussed in the related work section.**
>
> Hi Guangxiang,
>
> Thank you for letting us know about your work. Congratulations on the AAAI2022 acceptance.
>
> > The additional polynomial loss (e.g. 1−p) function has been proposed in our previous paper
>
> We will add your concurrent paper to the related work section. Our work is motivated by representing loss functions in $(1-P)^j$ bases, see Fig 1b, which is conceptually quite different from the references you provided. Again, it is no secret that loss function can be expanded in $(1-P)^j$ bases and modified by adjusting the coefficients of $(1-P)^j$,  as we have discussed in our related work section, e.g., [3][4] (published in 2020).
>
> >The best choice in this paper (i.e. 1−p) is a moderate choice in our paper
>
> Just to clarify: $(1-P)$ is not the best choice in our paper, e.g., $5(1-P)$ is optimal for ENetV2. In fact, adding $(1-P)$ sometimes hurts model performance, where a negative coefficient $-(1-P)$ is optimal, e.g., Mask R-CNN.
>
> One of the main points of our paper is that adjusting the coefficient $\epsilon_1$ of $(1-P)$ is important. Simply adding $(1-P)$ is suboptimal (Fig 3a, 5a, 6a).
>
> To be more specific, we focus on adjusting the coefficient of the leading polynomial because
> 1. The leading polynomial $(1-P)$ contributes a large fraction of the training gradient (Fig 3b)
> 2. The coefficient of the leading polynomial $(1+\epsilon_1)$  controls the confidence of the model predictions (Fig 5b, 6b).
>
> That being said, as stated in the conclusion, we agree that there are better losses compared to the simplest Poly-1 loss formulation.
>
> [3] Lei Feng, Senlin Shu, Zhuoyi Lin, Fengmao Lv, Li Li, and Bo An. Can cross entropy loss be robust to label noise. In Proceedings of the 29th International Joint Conferences on Artificial Intelligence, pp. 2206–2212, 2020.
>
> [4] Santiago Gonzalez and Risto Miikkulainen. Optimizing loss functions through multivariate taylor polynomial parameterization. arXiv preprint arXiv:2002.00059, 2020b.

---

> > ### Public Comment · ~Guangxiang_Zhao3 · 2022-04-13
> > **Your clarification is detailed and makes sense.**
> >
> > Thanks for your quick response, we appreciate it. We are happy to learn that other papers discussed "(1-P)^j ", and we agree that your work focus on different aspects.  We will also cite you in the next arxiv revision.
> > BTW，the finding that -(1-p) is optimal in Mask R-CNN is intriguing.
> >
> > The discussion of coefficients of polynomial loss is very meaningful. We are also intrigued by that.
> >
> > We did experiments of coefficients in Encouraging Loss, the difference is that we do not need a large coefficient for low p since we demonstrate that reviving the learning of well-classified examples (examples with high p) has theoretical benefits including enlarging margin.
> >
> > To be specific, we early end the log(1-p) curve at point LE and replace the log curve with the tangent of log(1-p)  at the LE. The maximum deviation of additional loss is 1/(1-LE).   E.g.,1 for LE=0, 2 for LE=0.5, 3 for LE=0.66, 4 for LE=0.75. The comparisons of results are in Figure 4.  But it is slightly different from your experiments,  only for LE=0, the coefficient is consistent for P(i.e., Loss=-logp-p).

---

### Decision · Program_Chairs · 2022-01-20

**Decision:**

Accept (Poster)

**Comment:**

Realizing the fact that cross-entropy loss and focal loss are widely used for training deep learning models but mathematical understanding and exploration for such losses are lacking, the authors propose a simple framework named PolyLoss to express the loss function as a linear combination of polynomial functions.

In this framework, the aforementioned cross-entropy loss and focal loss are the special cases of PolyLoss by easily adjusting the importance of different polynomial bases depending on the targeting tasks and datasets. The final version of PolyLoss, Poly-1 formulation, is simple with one line of code and an extra hyperparameter but outperforms the cross-entropy loss and focal loss on 2D image classification, instance segmentation, object detection, and 3D object detection tasks, sometimes by a large margin.

This paper is well-motivated by a novel perspective of polynomial expansion. The proposed method is novel, simple to implement, and effective in practice. The authors have a deep and thorough discussion with reviewers, and most concerns were well addressed. After rebuttal and discussion, reviewers increased their scores, and all agreed with acceptance. AC checked the paper and all relevant information, and found sufficient ground for acceptance.